# Production of Reactive Oxygen Species by Epicardial Adipocytes Is Associated with an Increase in Postprandial Glycemia, Postprandial Insulin, and a Decrease in Serum Adiponectin in Patients with Severe Coronary Atherosclerosis

**DOI:** 10.3390/biomedicines10082054

**Published:** 2022-08-22

**Authors:** Natalia V. Naryzhnaya, Olga A. Koshelskaya, Irina V. Kologrivova, Tatiana E. Suslova, Olga A. Kharitonova, Sergey L. Andreev, Alexander S. Gorbunov, Boris K. Kurbatov, Alla A. Boshchenko

**Affiliations:** Cardiology Research Institute, Tomsk National Research Medical Center, Russian Academy of Sciences, 111a Kievskaya Str., 634012 Tomsk, Russia

**Keywords:** reactive oxygen species, adipocytes, epicardial adipose tissue, postprandial glycemia, postprandial insulin, adiponectin, leptin, coronary atherosclerosis

## Abstract

**Purpose.** This work investigates the relations between the production of reactive oxygen species (ROS) by epicardial adipose tissue (EAT) adipocytes and parameters of glucose/insulin metabolism, circulating adipokines levels, and severity of coronary atherosclerosis in patients with coronary artery disease (CAD); establishing significant determinants describing changes in ROS EAT in this category of patients. **Material and methods.** This study included 19 patients (14 men and 5 women, 53–72 y.o., 6 patients with diabetes mellitus type 2; 5 patients with prediabetes), with CAD, who underwent coronary artery bypass graft surgery. EAT adipocytes were isolated by the enzymatic method from intraoperative explants obtained during coronary artery bypass grafting. The size of EAT adipocytes and ROS level were determined. **Results.** The production of ROS by EAT adipocytes demonstrated a direct correlation with the level of postprandial glycemia (rs = 0.62, *p* < 0.05), and an inverse correlation with serum adiponectin (rs = −0.50, *p* = 0.026), but not with general and abdominal obesity, EAT thickness, and dyslipidemia. Regression analysis demonstrated that the increase in ROS of EAT adipocytes occurs due to the interaction of the following factors: postprandial glycemia (β = 0.95), postprandial insulin (β = 0.24), and reduced serum adiponectin (β = −0.20). EAT adipocytes in patients with diabetes and prediabetes manifested higher ROS production than in patients with normoglycemia. Although there was no correlation between the production of ROS by EAT adipocytes and Gensini score in the total group of patients, higher rates of oxidative stress were observed in EAT adipocytes from patients with a Gensini score greater than median Gensini score values (≥70.55 points, Gr.B), compared to patients with less severe coronary atherosclerosis (<70.55 points, Gr.A). Of note, the frequency of patients with diabetes and prediabetes was higher among the patients with the most severe coronary atherosclerosis (Gr.B) than in the Gr.A. **Conclusions.** Our data have demonstrated for the first time that systemic impairments of glucose/insulin metabolism and a decrease in serum adiponectin are significant independent determinants of oxidative stress intensity in EAT adipocytes in patients with severe coronary atherosclerosis. The possible input of the interplay between oxidative stress in EAT adipocytes and metabolic disturbances to the severity of coronary atherosclerosis requires further investigation.

## 1. Introduction

Reactive oxygen species and redox signaling are important regulators of cellular functions in physiological conditions [1]. In adipocytes, ROS are involved in the regulation of lipolysis [2], transcription [3], homeostatic intracellular signaling, and the functioning of key intracellular signaling pathways [4], and provide regular renewal of adipocytes, taking part in the processes of differentiation of multipotent mesenchymal stem cells into mature adipocytes [5]. In physiological conditions, ROS are involved in intracellular signal transduction in response to insulin reception. For example, hydrogen peroxide mediates the action of insulin on adipocytes, which leads to a rapid translocation of glucose transporters, an increase in glucose uptake [6], and lipid synthesis [7], while the rate of lipolysis decreases [8]. In a study by Loh et al. (2009), ROS are also reported to increase cell sensitivity to insulin [9].

However, excessive production of ROS, known as adipocyte oxidative stress, characterized by the excessive formation of reactive oxygen species and a decrease in antioxidant protection, is a pathological process known to be a significant factor in cardiovascular pathology associated with metabolic disorders, including obesity [10,11,12,13,14,15]. Many clinical and experimental studies have demonstrated that obesity contributes to the development of systemic oxidative stress [11,12,16]; however, there are only a few studies evaluating the production of reactive oxygen species by adipocytes of adipose tissue [17,18,19,20,21].

In experimental studies, it was found that a significant factor in the increase in oxidative stress in adipocytes is the growth of glycemia and insulin resistance [11,18,19,22]. It has been established that a significant increase in ROS production in isolated adipocytes of mice with metabolic syndrome occurs at a high level of glucose in their incubation medium [19]. Adipocytes of visceral adipose tissue from rats and mice with hyperglycemia developed on a high-fat diet were characterized by both an increase in lipid peroxidation and a decrease in antioxidant enzymes (catalase, SOD, and glutathione peroxidase) activity [23,24,25]. Similar results were demonstrated in a model of genetically determined obesity in mice of the KK-A^y^ line [11]. Previously, hyperglycemia has also been reported to promote the superoxide anion production by activating a metabolic pathway within the cell that includes diacylglycerol, protein kinase C, and NADPH oxidase (the so-called “dangerous metabolic pathway in diabetes”) [26]. Subsequently, superoxide anion, spontaneously or under the influence of superoxide dismutase, turns into hydrogen peroxide, acquiring the ability to penetrate the cell membrane, and is released into the environment, contributing to an increase in systemic oxidative stress [27].

There is evidence that ROS production in visceral fat adipocytes of patients with metabolic syndrome is five-fold higher than that in the control group [18]. The authors conclude that metabolic syndrome contributes to oxidative stress in adipose tissue, mainly due to the activation of ROS production by adipocytes. To date, the mechanisms of adipogenic oxidative stress in patients with cardiometabolic diseases have not been fully established; the data on the ability of epicardial adipocytes to produce reactive oxygen species in patients with coronary heart disease are extremely limited [20]. Recently, in our pilot study, we have demonstrated that the production of ROS by EAT adipocytes in CAD patients with severe coronary atherosclerosis is directly related to the level of postprandial glycemia [28]. It has been demonstrated that EAT has a greater potential for ROS production compared to subcutaneous adipose tissue, due to a higher expression of the NADPH components gp91phox and p47phox [29]. In addition, patients with coronary artery disease have an increase in the relative content of catalase in EAT, which, possibly, has a compensatory nature due to an acceleration of ROS production [30].

At the same time, the association of oxidative stress in EAT adipocytes with disorders of carbohydrate metabolism, the adipokine profile, and the severity of coronary atherosclerosis in patients with coronary artery disease (CAD) has not yet been evaluated.

The purpose of this study was to investigate the relations between the production of reactive oxygen species (ROS) by epicardial adipose tissue (EAT) adipocytes and parameters of glucose/insulin metabolism, circulating adipokines levels, and severity of coronary atherosclerosis in patients with CAD; to establish significant determinants describing changes in ROS EAT in this category of patients.

## 2. Methods

The study was conducted following the Declaration of Helsinki of the World Medical Association “Ethical principles for conducting scientific medical research involving humans” as amended in 2000 and “Rules of Clinical Practice in the Russian Federation”, approved by the Order of the Ministry of Health of the Russian Federation on 19 June 2003 No. 266. The study’s protocol was approved by the Biomedical Ethics Committee of Cardiology Research Institute, Tomsk NRMC, protocol No. 210 from February 18, 2021. All subjects provided their written informed consent before being enrolled in the study. All patients received the optimal therapy.

### 2.1. Study Population

The clinical characteristics of patients are presented in Table 1. This pilot study included 19 patients (14 men and 5 women; 53–72 y.o.) with coronary artery disease scheduled for coronary artery bypass grafting.

The exclusion criteria were acute cardiovascular events within the last 6 months; active, ongoing inflammatory diseases other than atherosclerosis or elevation of high-sensitive C-reactive protein (hsCRP) ≥ 10 mg/L; history of active, ongoing, or recurrent infections; chronic kidney disease class above C3b; cancer, hematological, and autoimmune diseases, as well as a change in body weight of more than 3% in the previous 3 months.

Anthropometric measurements were performed to assess total obesity according to the level of body mass index (BMI) and abdominal obesity according to the size of the waist circumference, hip circumference, and waist-to-hip ratio (WHR). Body composition was assessed by Bioelectrical Impedance Analysis.

EAT thickness was measured on the free wall of the right ventricle in a still image at the end-diastole on the parasternal long-axis view in 3 cardiac cycles at the end of the systole [31]. EAT thickness was measured at the point of perpendicular orientation of the ultrasound beam on the free wall of the right ventricle, using the aortic annulus as an anatomic landmark. The thickest point of EAT was measured in each cycle. The EAT thickness was calculated as an average value from echocardiographic views in 3 cardiac cycles.

All patients underwent selective coronary angiography on a Cardioscop-V angiographic complex and Digitron-3NAC computer system, Siemens (Germany). The severity of CAD was assessed by the value of Gensini score [32]. Since the severity of coronary atherosclerosis differed significantly in men and women (median values 88 points and 53 points for men and women, respectively), this parameter was adjusted by sex.

### 2.2. Biochemical Study

The content of leptin (Mediagnost, Reutlingen, Germany), adiponectin (Assaypro, St. Charles, MO, USA), insulin (AccuBind, Diagnostic System Laboratories, Lake Forest, CA, USA), and hsCRP (Biomerica, Irvine, CA, USA) were determined in blood serum by enzyme-linked immunosorbent assay (ELISA). An adjustment of differences in the leptin level for gender and BMI was carried out. The level of adiponectin did not depend on gender; this parameter was adjusted only for BMI.

The level of glucose was detected by hexokinase assay (EKF diagnostic, Leipzig, Germany). The Enzyme colorimetric method was used to estimate the serum concentration of total cholesterol, triacylglycerol, and high-density lipoprotein (HDL) cholesterol (Diakon, Pushchino, Russia). Concentration of low-density lipoprotein (LDL) cholesterol was calculated using the formula [LDL] = [Total cholesterol] − [Triacylglycerol (TG)] − [HDL].

The diagnosis of diabetes was based on generally accepted European guidelines. The term “prediabetes” was used to identify conditions of impaired fasting glucose (IFG, fasting plasma glucose between 6.1–6.9 mM and with plasma glucose after OGTT 2-h < 7.8 mM) or impaired glucose tolerance (IGT, fasting plasma glucose < 7.0 mM and OGTT 2-h glucose 7.9–11.1 mM) or a combination of both [33].

### 2.3. Adipose Tissue Explants

The explants of EAT weighing 0.2–1 g obtained during the CABG surgery comprised the material for the study, as we reported earlier [34]. Shortly, epicardial fat tissue explants were taken from the tissue surrounding the proximal parts of the right coronary artery. Adipose tissue cells were isolated enzymatically in type I collagenase sterile solution (PanEco, Moscow, Russia) at 1 mg/mL in Krebs-Ringer buffer. The cell suspension was filtered through a nylon filter (Falcon™ Cell strainer, pore diameter 100 μm) and washed. The number and size of the obtained adipocytes were counted using light microscopy (Axio Observer.Z1, Carl Zeiss Surgical GmbH, Oberkochen, Germany). Cells were stained with Hoechst 33,342 (5 μg/mL, stains nucleus of viable cells) and propidium iodide (10 μg/mL, Sigma-Aldrich, St. Louis, MO, USA, stains nucleus of dead cells) to distinguish viable cells from dead cells (Figure 1) [35]. Samples with viability lower than 95% were excluded from the study. The remaining samples did not differ significantly in the percentage of viable cells.

To measure the level of reactive oxygen species, 200 μL adipocytes in Krebs-Ringer buffer (1.25 × 10^6^ cells/mL) were added to the two wells of a 96-well plate (500,000 cells per well) and were incubated for 30 min in the presence of 125 μM 2,3-dihydrodichlorofluorescein diacetate (DCF-DA) in a microplate reader (INFINITE 200M; Tecan, Grödig, Austria) at 37 °C. The fluorescence of DCF was measured at a wavelength of λ_ex_ = 500, λ_em_ = 530. The accumulation of reactive oxygen species by adipocytes was verified microscopically (Figure 1).

Data analysis was performed using STATISTICA 13.0 software (StatSoft Inc., Tulsa, OK, USA). The normality of the distribution of sample data was verified by the Shapiro–Wilk test. Data were presented as the median and interquartile range (Q_25th_–Q_75th_) when distribution was different from normal. Categorical data were described by absolute (n) and relative (%) frequencies. To identify statistically significant differences in independent groups, the Mann–Whitney test was used for quantitative parameters and Pearson’s chi-square test was used for categorical parameters. The study of correlations between variables was carried out using the Spearman correlation coefficient. We also investigated the associations between carbohydrate metabolism disturbances, adiponectin level, and ROS EAT production using the multifactor linear and nonlinear regression analysis. All the statistical hypotheses were accepted when the significance level was less than *p* < 0.05.

## 3. Results

Table 2 presents the biochemical characteristics of the patients included in the study.

We did not detect linear relationships of EAT ROS with BMI, WHR, basal glycemia, and insulinemia, as well as parameters of blood lipid transport function.

Correlation analysis revealed three possible non-collinear markers of increased ROS in EAT: postprandial glucose (r_s_ = 0.616, *p* = 0.013, Figure 2A), postprandial insulin (r_s_ = 0.061, *p* = 0.126, (Figure 3A), and adiponectin (r_s_ = −0.503, *p* = 0.026, Figure 4A).

Variables EAT ROS and postprandial glucose had a distribution close to normal (according to the Shapiro–Wilk test *p* = 0.057 and *p* = 0.078, respectively), with the relationship between EAT ROS and postprandial glucose being the closest to linear. However, the possibility to create a simple linear regression model of postprandial glucose on EAT ROS was limited due to the high heteroscedasticity of the sample data and the small sample size (*n* = 19). The empirical correlation ratio of the degree of strength of the relationship between ROS EAT and postprandial glucose was ρ = 0.693, 95% CI (0.586; 0.767) (Figure 2B).

Even though we did not reveal the statistically significant linear relationship between EAT ROS increase with elevated postprandial insulin (Figure 3A), clinical considerations prompted us to explore the presence and form of relationship between EAT ROS and postprandial insulin level. The relationship between these variables appeared to be highly nonlinear and was approximated by a third-degree polynomial (Figure 3B). The correlation ratio of the degree of the relationship strength between ROS EAT and postprandial insulin constituted ρ = 0.613, 95% CI (0.581; 0.692) (Figure 3B).

The relationship between EAT ROS and adiponectin also appeared to be close to linear (Figure 4). However, as in the case of postprandial glucose, it was not possible to build a linear regression model with normally distributed residuals due to the lack of normal distribution of the adiponectin variable (according to the Shapiro–Wilk test *p* = 0.00011) and data heteroscedasticity. The empirical correlation ratio of the degree of strength of the relationship between ROS EAT and adiponectin constituted ρ = 0.726, 95% CI (0.511; 0.832).

Thus, we concluded that the variables postprandial glucose, adiponectin, and postprandial insulin could be chosen as screening markers of EAT ROS increase in patients with coronary atherosclerosis.

The direct correlation with serum leptin was not statistically significant.

A non-linear multivariate regression analysis was performed, which revealed that postprandial glycemia and postprandial insulinemia positively affected the production of ROS EAT in a complex way (Figure 5).

A statistically significant multiple linear regression model (without the intercept), which included postprandial glycemia, postprandial insulinemia, and serum adiponectin as determinants of the intensity of ROS EAT production, was also constructed (Table 3).
ROS EAT = 236.63 + 0.563 PPG + 0.357 PPI − 0.348 Adipo(1)

Note: PPG—postprandial glucose, mM; PPI—postprandial insulin, μU/mL; Adipo—serum adiponectin, adjusted to BMI, µg/mL. Analysis of the residuals of the model showed their normality.

Further, the patients were divided into two groups, depending on their glycemic state. The patients with normoglycemia constituted the first group; patients with diabetes and prediabetes (IFG, IGT, or a combination of both) constituted the second group. The levels of fasting, postprandial glycemia, and glycated hemoglobin were higher in the patients of the second group, as expected, while there were no intergroup differences in fasting and postprandial insulinemia, obesity measurements, lipid metabolism, and serum adipokines (Table 4).

The ROS production by EAT adipocytes was significantly higher in patients with diabetes and prediabetes than in patients with normoglycemia (*p* = 0.023) (Figure 6).

EAT thickness and EAT adipocyte size had no intergroup differences (Table 5). Even though we did not detect statistically significant differences between groups in respect of the Gensini score (possibly due to the small number of enrolled patients), its values were higher in Group 2 (Table 5).

We did not find a significant linear relationship between ROS EAT and the severity of atherosclerosis as assessed by the Gensini score (r_s_ = 0.44, *p* = 0.065) (Figure 7).

To perform an in-depth analysis of the involvement of EAT ROS in the modulation of the severity of coronary atherosclerosis, we divided the entire group of patients with coronary heart disease into two groups depending on the values of the Gensini score. After adjusting Gensini’s score to gender, its median constituted 70.55 points. Hence, patients with an adjusted Gensini score less than 70.55 points were referred to as group A, and those with a Gensini score equal to or exceeding 70.55 points were referred to as group B.

The median Gensini score was 48 points in group A and 148.6 points in group B (Table 6). The production of reactive oxygen species by EAT adipocytes was more intense in patients with Gensini scores over 70.55 points, compared to patients with Gensini scores less than 70.55 points (Figure 8).

Group B primarily consisted of diabetic and prediabetic patients, who demonstrated higher levels of leptin and lower values of the adiponectin-to-leptin ratio (Appendix A). We did not reveal any differences between the groups in terms of obesity, lipid transport function, as well as the morphological characteristics of EAT (Table 6 and Appendix A).

## 4. Discussion

The association of oxidative stress in the EAT adipocyte with impaired carbohydrate metabolism in patients with coronary heart disease has not been reported in clinical studies to date. Our study is the first one to establish a direct correlation between ROS production in EAT and multiple metabolic parameters in patients with coronary heart disease. It should be noted that in this small sample of patients with coronary artery disease, the significant factors of oxidative stress in EAT adipocytes appeared to be not just manifest disorders of carbohydrate metabolism (diabetes mellitus type 2), but primarily the state of prediabetes, which encompassed latent disorders of glucose/insulin metabolism and was associated with postprandial hyperglycemia and hyperinsulinemia. Thus, we have demonstrated that the production of ROS by EAT adipocytes remains at a low level only in patients with preserved carbohydrate metabolism, while patients with diabetes and prediabetes have a higher rate of oxidative stress in these cells. The presented results indicate that insulin, along with glycemia, acts as an additional factor increasing oxidative stress in EAT adipocytes. Our earlier data evidenced the insulin-mediated dysregulation of EAT adipocytes. The relationships between serum insulin levels (both postprandial and basal) and insulin resistance of EAT adipocytes were demonstrated in patients with coronary heart disease [28]. In the same study, we demonstrated that insulin resistance of the EAT adipocyte in this category of patients was closely associated with the severity of coronary atherosclerosis [28]. These data support the current view that, in contrast to the physiological role of ROS in insulin signaling, ROS overproduction promotes insulin resistance and impairs glucose uptake in adipocytes [36,37].

Metabolic disorders observed in obesity are closely associated with an imbalance of the blood adipokines profile, an increase in the activity of pro-inflammatory adipokines, and a decrease in the content of adiponectin [38,39,40,41].

It is well known that there is a reduced level of adiponectin in patients with coronary atherosclerosis [42] as well as in the combination of CAD with obesity [43]; it was the establishment that adiponectin is an independent risk factor of coronary artery disease occurrence, but not its advancement [44]. However, to date, a potential relationship between blood adiponectin and oxidative stress in EAT adipocytes in patients with coronary atherosclerosis has not been reported. Our study revealed an inverse relationship between serum adiponectin and ROS of EAT adipocytes with r_s_ = −0.50. Moreover, the multivariate regression analysis performed in our study demonstrated that the decrease in adiponectin acts as a factor that enhances the pro-oxidative effect of carbohydrate metabolism disorders on EAT.

The mechanisms of the inhibitory effect of adiponectin on the oxidative stress of adipocytes are mainly mediated through its anti-inflammatory activity [38,45], and its effect on ROS production by mitochondria [42].

The beneficial effects of adiponectin on the intracellular mechanisms of ROS regulation through nitric oxide synthase and NADPH+-oxidase should also be emphasized [46,47]. In addition, it was found that adiponectin increases the sensitivity of cells to insulin, possibly through the realization of its anti-inflammatory effect [48]. Thus, a decrease in the content of adiponectin may be associated with the development of insulin resistance in adipocytes and is of relevance due to the initially lower production of adiponectin by EAT adipocytes compared to subcutaneous adipose tissue [49]. Moreover, the presence of coronary artery disease in patients leads to an even more pronounced deficiency of adiponectin in EAT [50]. It should be noted that the relationship between the production of ROS EAT and the content of serum leptin was not established in our study.

What are the consequences of increased oxidative stress in adipocytes? In recent studies, the significant role of oxidative stress of adipocytes in the formation of functional and structural cardiac disorders, the consequence of which is the development of heart failure, has been discussed [10]. It was also reported that the consequence of oxidative stress in adipocyte is represented by the change in its adipokine profile towards the release of pro-inflammatory adipokines [11,51]. It has been demonstrated that the secretion of pro-inflammatory adipokines by EAT adipocytes contributes to coronary endothelial damage in patients with coronary artery disease and obesity and can be prevented by adiponectin [52]. In this article, it was demonstrated that the patient’s metabolic status and obesity amplify the pro-inflammatory effect of adipokines on the vascular endothelium.

The relationships between oxidative stress of EAT adipocytes, an increase in the secretion of pro-inflammatory adipokines, and a decrease in the release of adiponectin with the incidence of cardiac arrhythmias, are discussed [53]. The experimental study demonstrated that metformin amendment of EAT adipocytes’ adiponectin secretion prevented pathological atrial remodeling and weakened the vulnerability to atrial fibrillation induced by atrial pacing [54].

The association of oxidative stress of adipocytes of perivascular adipose tissue with dysregulation of vascular tone and homeostasis has been established [55].

It is known that both an increase in postprandial glycemia and oxidative stress of the EAT adipocyte may be associated with pro-inflammatory status [38]. Meanwhile, chronic low-grade inflammation is a significant pathogenetic factor in atherosclerosis and coronary artery disease [56].

However, the relationship between oxidative stress in the EAT adipocyte and coronary atherosclerosis has been reported only in one study. It was found that patients with coronary atherosclerosis have a higher level of EAT ROS than patients without coronary pathology [20]. However, there are no data on a potential relationship between ROS production in EAT and the severity of coronary atherosclerosis. In our study, we failed to demonstrate a significant linear relationship between the production of ROS by EAT adipocytes and the severity of coronary atherosclerosis, assessed by the Gensini score index. It may be due to both the insufficient predictive power of the Gensini score index and the small sample size. At the same time, we found that in the group of patients with more severe coronary atherosclerosis, the production of ROS by EAT adipocytes is higher. Since group B primarily consisted of diabetic and prediabetic patients, one cannot exclude that it is an interplay between metabolic impairments and increased EAT ROS, but not oxidative stress alone, that was associated with the severity of atherosclerosis. The division into groups was arbitrary and does not allow us to consider the Gensini score = 70.55 points as the threshold for determining oxidative stress. Moreover, the cross-sectional design of the study does not allow us to establish the consecutive links between EAT oxidative stress and the progression of atherosclerosis. Despite the predominant opinion about the direction of humoral reactions from EAT to the myocardium, the possibility of increased oxidative stress of EAT as a consequence of coronary circulation disorders and the humoral effect of ischemic myocardium on adipose tissue cannot be ruled out. Further prospective studies aiming to elucidate the involvement of EAT ROS in atherogenesis are required.

In further clinical studies, it is important to evaluate the potential beneficial effect of postprandial hyperglycemia correction with hypoglycemic therapy on ROS production by EAT adipocytes in patients with documented atherosclerosis, regardless of the presence or absence of type 2 diabetes mellitus, as well as to investigate the nature of changes in glucose/insulin metabolism and ROS production by adipocytes under the influence of antioxidant treatment.

## 5. Conclusions

It has been demonstrated for the first time that oxidative stress of EAT adipocytes in patients with coronary artery disease receiving conventional therapy is associated with the complex effect of low circulating adiponectin and elevated postprandial levels of glycemia and insulinemia, but not with parameters of general, abdominal obesity and dyslipidemia. The highest intensity of ROS production in EAT adipocytes in patients with coronary heart disease occurs in patients with manifest and latent disorders of carbohydrate metabolism. Further prospective studies are required to confirm the role of oxidative stress of EAT adipocytes in mediating the relationships between impaired glucose/insulin metabolism, adipokines profile, and progression of coronary atherosclerosis.

## 6. Limitations

The main limitations of our study include the relatively low number of the recruited patients and its cross-sectional design. In our study, no relationship was found between the diagnosis of diabetes mellitus and adipocyte ROS. This may be due to the presence of latent disorders of glucose/insulin metabolism in some patients and the use of hypoglycemic drugs in all diabetic patients. In addition, due to the small sample size, we did not find statistically significant differences in the use of drug therapy in groups 1 and 2. However, these differences for a larger sample may be important.

## Abbreviation

BMI, body mass index; CAD, coronary artery disease; DCF-DA 2,3-dihydrodichlorofluorescein diacetate; EAT, epicardial adipose tissue; ELISA, enzyme-linked immunosorbent assay; HDL, high-density lipoproteins; IFG, impaired fasting glucose; IGT, impaired glucose tolerance; OGTT, oral glucose tolerance test; LDL, low-density lipoproteins; ROS, reactive oxygen spices; WHR, waist-to-hip ratio.

## Figures and Tables

**Figure 1 biomedicines-10-02054-f001:**
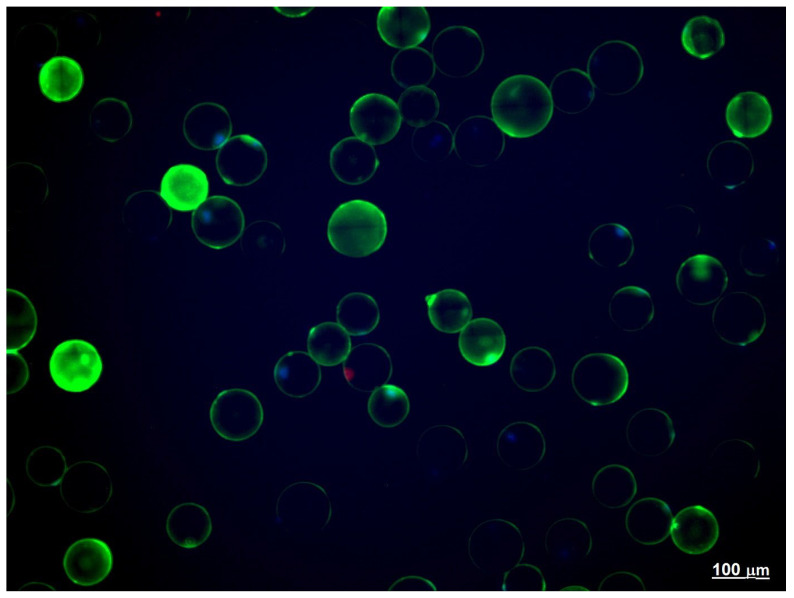
Accumulation of reactive oxygen species (ROS) and viability of adipocytes in epicardial adipose tissue (EAT) culture. Notes: Fluorescence staining. Dyes: green—2,3-dihydrodichlorofluorescein (ROS), red—propidium iodide (dead cells), blue—Hoechst 33,342 (viable cells). Magnification ×200.

**Figure 2 biomedicines-10-02054-f002:**
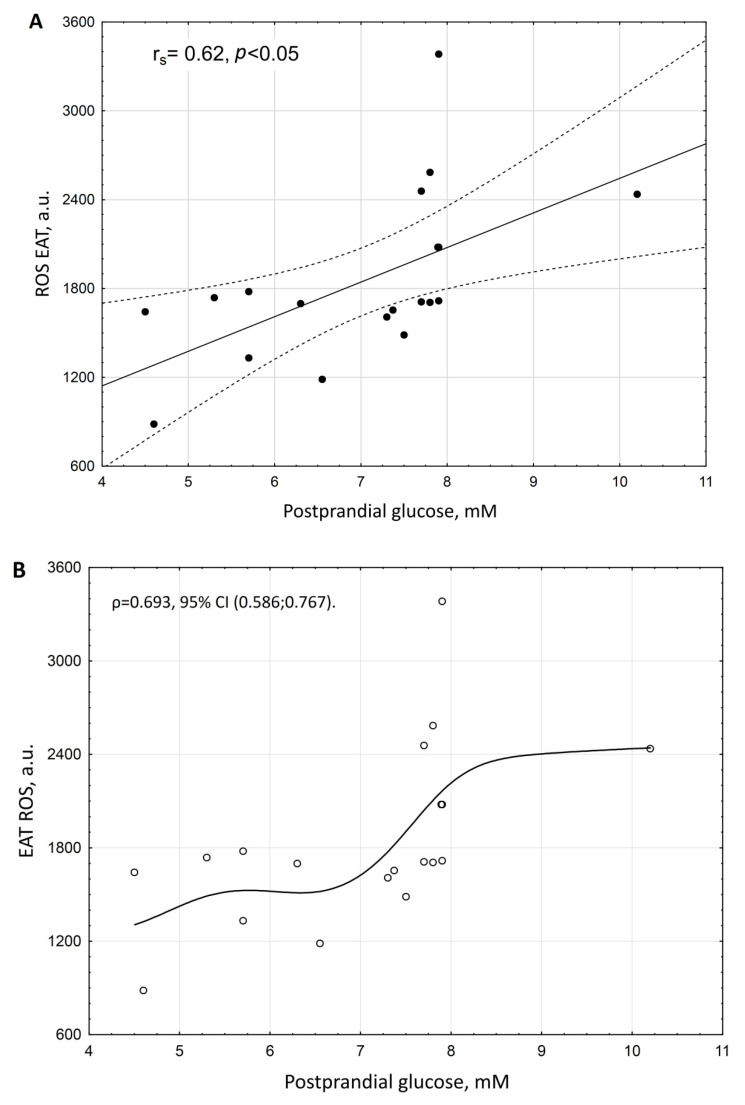
Scattering diagram of ROS production by EAT adipocytes and serum level of postprandial glucose in patients with coronary artery disease and coronary atherosclerosis. (**A**)—Spearman correlation coefficient (r_s_). (**B**)—empirical regression line.

**Figure 3 biomedicines-10-02054-f003:**
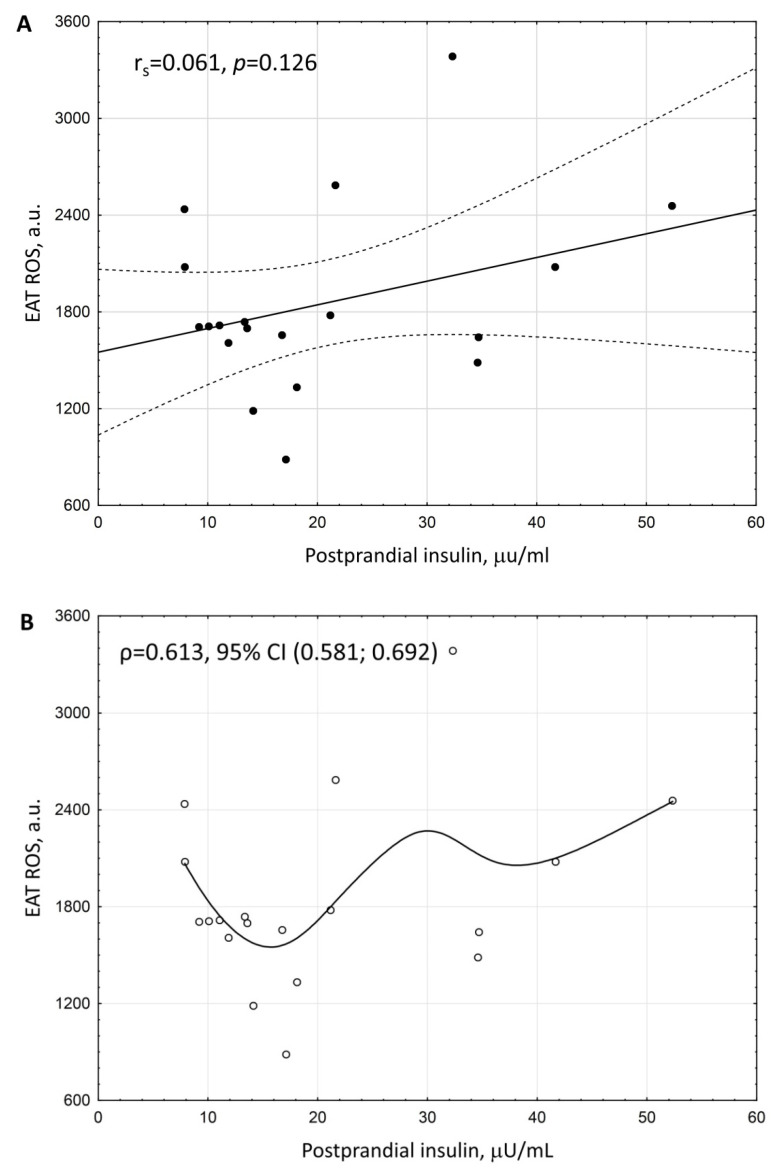
Scattering diagram of ROS production by EAT adipocytes and the serum level of postprandial insulin in patients with coronary artery disease and coronary atherosclerosis. (**A**)—Spearman correlation coefficient (r_s_). (**B**)—empirical regression line. Adiponectin level was adjusted to BMI.

**Figure 4 biomedicines-10-02054-f004:**
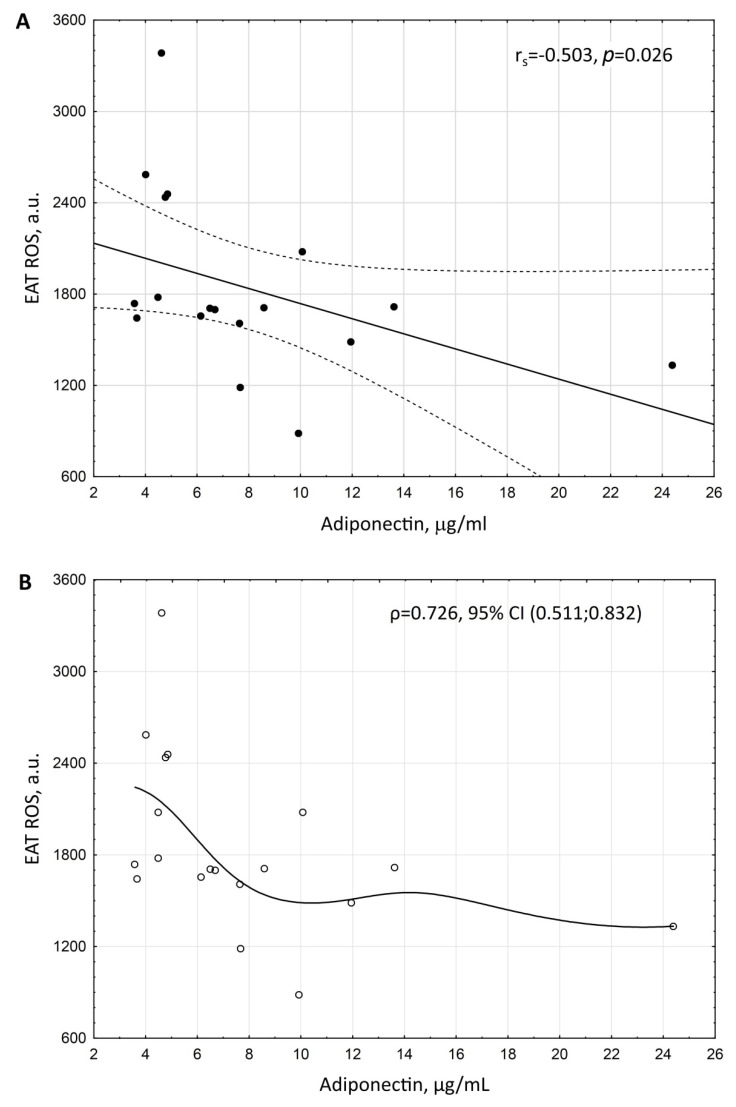
Scattering diagram of ROS production by EAT adipocytes and the serum level of adiponectin in patients with coronary artery disease and coronary atherosclerosis. (**A**)—Spearman correlation coefficient (r_s_). (**B**)—empirical regression line. Adiponectin level was adjusted to BMI.

**Figure 5 biomedicines-10-02054-f005:**
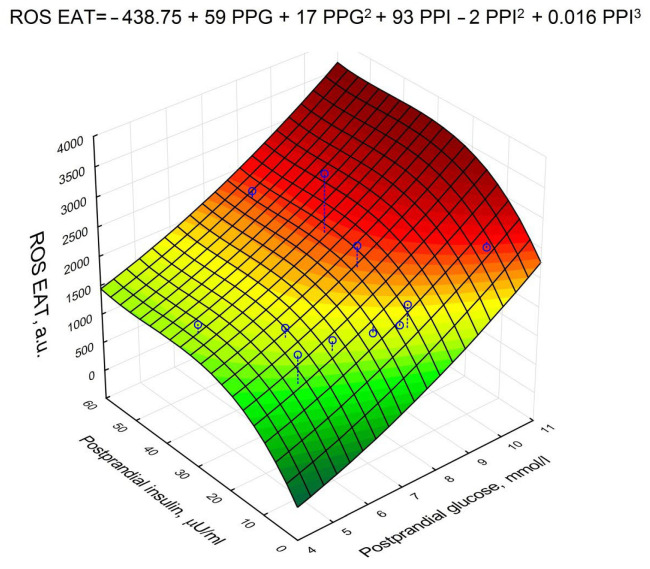
Response surface of the regression model of postprandial glycemia and postprandial insulinemia on ROS EAT.

**Figure 6 biomedicines-10-02054-f006:**
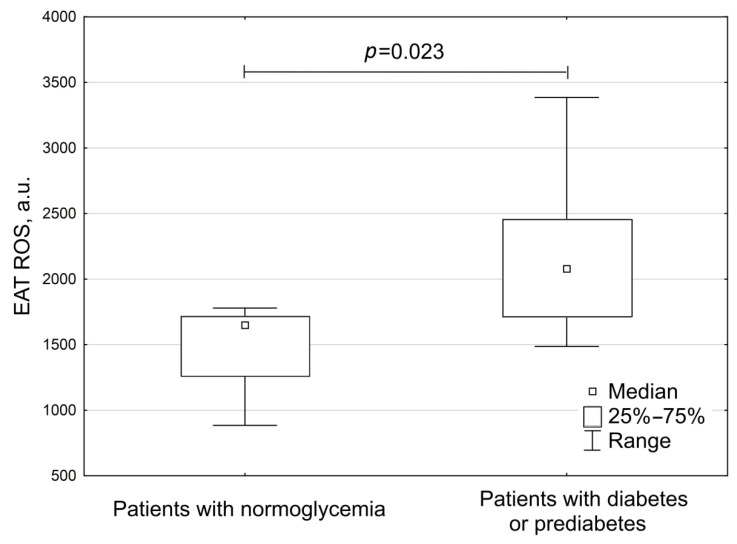
The production of reactive oxygen species by EAT in patients with coronary artery disease, depending on the glycemic states.

**Figure 7 biomedicines-10-02054-f007:**
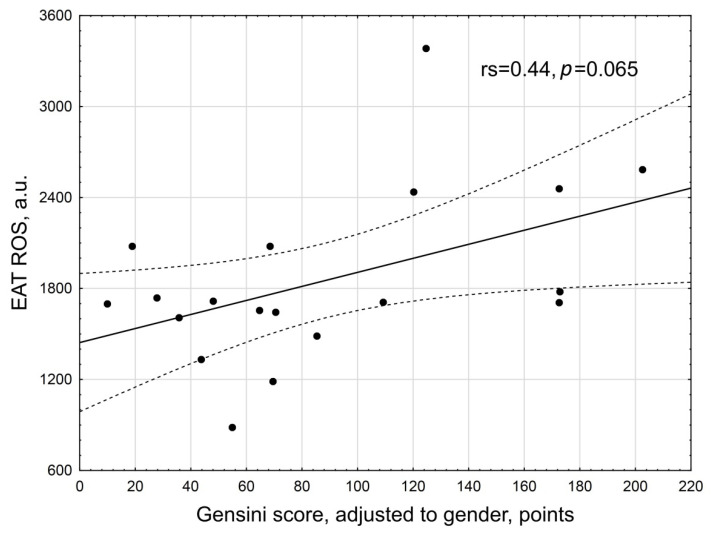
Scattering diagram of Gensini score related to the ROS production by EAT adipocytes in the general group of CAD patients. Note: r_s_—Spearman correlation coefficient.

**Figure 8 biomedicines-10-02054-f008:**
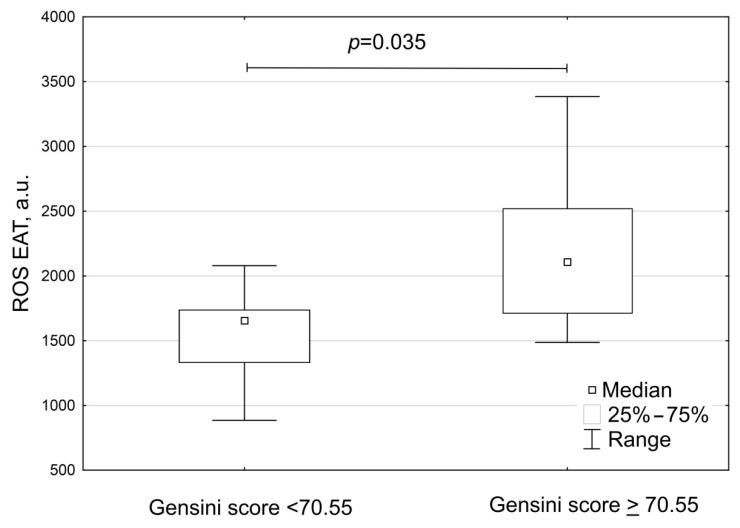
The production of reactive oxygen species by EAT in patients with coronary artery disease, depending on the coronary atherosclerosis severity: less than 70.55 points (*n* = 11) and more than 70.55 points (*n* = 8), Gensini score. Note: Gensini score was adjusted to gender.

**Table 1 biomedicines-10-02054-t001:** Clinical characteristics of patients (*n* = 19).

Parameters.	
Gender (m/f), *n* (%)	14 (74)/5 (26)
Age, years, Me (max-min)	62 (53–72)
History of myocardial infarction, *n* (%)	7 (37)
Hypertension, *n* (%)	19 (100)
Diabetes mellitus, *n* (%)	6 (31.6)
Prediabetes, *n* (%)	5 (26.3)
Patients with normoglycemia, *n* (%)	8 (42.1)
Duration of hypertension, years	14 (10; 20)
Duration of CAD, years	2 (1; 7)
Systolic blood pressure, mmHg	135 (123.5; 144)
Diastolic blood pressure, mmHg	80 (70; 85)
Smoking, *n* (%)	7 (37)
Body mass index, kg/m^2^	30.1 (27.4; 33.3)
Obesity, *n* (%)	7 (37)
Waist circumference, cm	105 (97; 114)
Waist-to-hip ratio	0.98 (0.93; 1.09)
Fat-free mass, kg	57.90 (47.05; 62.05)
Fat mass, adjusted to BMI, kg	33.50 (28.95; 39.05)
Skeletal muscle mass, kg	26.30 (20.20; 28.50)
Gensini score, points	70.0 (28; 110)
Gensini score adjusted to gender, points	70.55 (48; 149)
EAT thickness, mm	4.65 (4.30; 5.60)
EAT adipocyte size, mm	87.05 (84.82; 89.52)
Metformin, *n* (%)	4 (21)
Angiotensin-converting enzyme inhibitors, *n* (%)	8 (42)
Angiotensin receptor blockers, *n* (%)	7 (37)
Calcium channels antagonists, *n* (%)	10 (53)
Diuretics, *n* (%)	17 (90)
Statins, *n* (%)	7 (37)

Note: data are presented as median (Me) and interquartile range (Q_25%_; Q_75%_), *n* (%).

**Table 2 biomedicines-10-02054-t002:** Biochemical characteristics of patients.

Parameters	The General Group of Patients, *n* = 19
Fasting glycemia, mmol/L	5.7 (5.15; 6.05)
Postprandial glycemia, mmol/L	7.5 (5.7; 7.89)
Fasting insulin, μU/mL	5.6 (4.65; 8.60)
Postprandial insulin, μU/mL	16.76 (11.06; 32.31)
HbA1c, %	6.36 (5.57; 7.04)
Total cholesterol, mmol/L	3.74 (3.01; 4.31)
Triacylglycerol, mmol/L	1.31 (1.05; 1.45)
HDL, mmol/L	1.04 (0.95; 1.23)
LDL, mmol/L	1.95 (1.60; 2.42)
Serum adiponectin, adjusted to BMI, μg/mL	6.36 (5.12; 10.44)
Serum leptin, adjusted to BMI and gender, ng/mL	18.18 (11.98; 27.79)
Adiponectin, adjusted to BMI/leptin, adjusted to BMI and gender	0.41 (0.27; 0.64)
hsCRP, mg/L	2.99 (0.96; 6.72)

**Table 3 biomedicines-10-02054-t003:** Regression coefficients of links between postprandial glycemia, postprandial insulinemia, serum adiponectin, and ROS EAT.

Parameter	Beta	*p*
Postprandial glucose	0.950704	<0.001
Postprandial insulin	0.236687	0.012
Serum adiponectin *	−0.203832	0.04

Notes: * Serum adiponectin was adjusted to BMI.

**Table 4 biomedicines-10-02054-t004:** Comparison of clinical and biochemical characteristics of patients depending on glycemic status.

Parameters	Group 1 Patients with Normoglycemia (*n* = 8)	Group 2 Patients with Diabetes or Prediabetes (*n* = 11)	*p*
Gender (m/f), *n* (%)	6 (75)/2 (25)	8 (73)/3 (27)	>0.05
Age, years, Me (max-min)	62 (53–71)	62 (53–72)	>0.05
BMI, kg/m^2^	29.1 (27.4; 30.8)	31.3 (26.3; 36.8)	0.21
Waist circumference, cm	102 (94.5; 107)	109 (98; 118)	0.139
Waist-to-hip ratio	0.96 (0.92; 1.01)	1.02 (0.93; 1.11)	0.46
Fat-free mass, kg	52.30 (47.00; 58.30)	60.70 (56.70; 67.70)	0.22
Fat mass, adjusted to BMI, kg	34.40 (27.70; 40.00)	33.35 (30.20; 36.70)	0.93
Skeletal muscle mass, kg	23.35 (18.00; 27.20)	27.30 (25.80; 31.30)	0.29
Fasting glycemia, mmol/L	5.2 (4.95; 5.7)	6.0 (5.7; 8.0)	**0.01**
Postprandial glycemia, mmol/L	5.7 (4.95; 6.43)	7.8 (7.7; 7.9)	**<0.001**
Fasting insulin, μU/mL	6.0 (5.0; 8.0)	5.6 (4.5; 8.7)	0.98
Postprandial insulin, μU/mL	16.9 (13.9; 19.6)	11.9 (9.2; 34.6)	0.59
HbA1c, %	5.6 (5.5; 6.04)	6.8 (6.3; 7.3)	**0.028**
Total cholesterol, mmol/L	3.84 (3.13; 4.21)	3.74 (3.01; 4.72)	0.71
Triacylglycerols, mmol/L	1.14 (0.91; 1.31)	1.37 (1.09; 1.78)	0.1
HDL, mmol/L	1.04 (1.00; 1.18)	1.06 (0.80; 1.36)	0.96
LDL, mmol/L	2.06 (1.61; 2.56)	1.95 (1.60; 2.42)	0.9
Serum adiponectin, adjusted to BMI, μg/mL	7.14 (5.37; 9.68)	6.36 (5.11; 10.6)	0.96
Serum leptin, adjusted to BMI and to gender, ng/mL	17.40 (10.61; 25.14)	18.34 (13.89; 27.79)	0.5
Adiponectin, adjusted to BMI/leptin, adjusted to BMI and gender	0.41 (0.26; 0.68)	0.43 (0.27; 0.64)	0.82
Metformin, *n* (%)	0	4 (36)	>0.05
Angiotensin-converting enzyme inhibitors, *n* (%)	2 (25)	6 (55)	>0.05
Angiotensin receptor blockers, *n* (%)	3 (38)	4 (36)	>0.05
Calcium channels antagonists, *n* (%)	3 (38)	7 (64)	>0.05
Diuretics, *n* (%)	7 (88)	10 (91)	>0.05
Statins, *n* (%)	2 (25)	5 (45)	>0.05

Notes: HDL, high-density lipoproteins; LDL, low-density lipoproteins.

**Table 5 biomedicines-10-02054-t005:** Comparison of coronary atherosclerosis severity and morphological parameters of epicardial adipose tissue of patients with different glycemic state.

Parameters	Group 1 Patients with Normoglycemia (*n* = 8)	Group 2 Patients with Diabetes or Prediabetes (*n* = 11)	*p*
Gensini score, points	38.50 (28.00; 70.50)	88.00 (36.00; 121.00)	0.18
Gensini score adjusted to gender, points	59.80 (35.75; 70.01)	109.24 (48.04; 172.55)	0.18
EAT thickness, mm	4.60 (4.10; 5.13)	4.65 (4.30; 5.60)	0.67
EAT adipocyte size, μm	83.24 (79.18; 89.19)	87.59 (86.37; 89.52)	0.11

**Table 6 biomedicines-10-02054-t006:** Comparison of coronary atherosclerosis severity and morphological parameters of epicardial adipose tissue in patients with Gensini score of more or less than 70.55 points.

Parameters	Group A Patients with Gensini Score < 70.55 (*n* = 11)	Group B Patients with Gensini ≥ 70.55 (*n* = 8)	*p*
Gensini score, points	33.0 (24.5; 69.0)	115.5 (88.0; 149.8)	**<0.001**
Gensini score adjusted to gender, points	48.0 (27.8; 68.5)	148.6 (114.7; 172.7)	**<0.001**
EAT thickness, mm	5.1 (4.4; 5.7)	4.3 (3.9; 4.8)	0.075
EAT adipocyte size, μm	85.5 (79.3; 89.1)	88.1 (86.5; 90.0)	0.12

Notes: Gensini score = 70.55 points—median value of Gensini score in the sample adjusted to gender.

## Data Availability

The data presented in this study are available on request from the corresponding author.

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
