# Peer review of "Production of Reactive Oxygen Species by Epicardial Adipocytes Is Associated with an Increase in Postprandial Glycemia, Postprandial Insulin, and a Decrease in Serum Adiponectin in Patients with Severe Coronary Atherosclerosis"

_biomedicines, 2022, doi:10.3390/biomedicines10082054_

Round 1

Reviewer 1 Report

The authors investigated the relations between the production of reactive oxygen species (ROS) by epicardial adipose tissue (EAT) adipocytes and parameters of glucose/insulin metabolism, circulating adipokines levels and severity of coronary atherosclerosis in 19 patients with coronary artery disease (CAD).  They found that the production of ROS by EAT adipocytes demonstrated a direct correlation with the level of postprandial glycemia. Their data have shown that systemic impairments of glucose/insulin metabolism and a decrease in serum adiponectin are significant independent determinants of oxidative stress intensity in EAT adipocytes in patients with severe coronary atherosclerosis.

This is a well-designed, nicely presented work presenting some interesting novel data on EAT adipocyte function.

Comments:

1.       Data on medication of the enrolled patients should be included. It is well known that some of the drugs used in CAD, including stating may significantly alter the levels of oxidative stress markers and adipokines. This could be also mentioned as a limitation of the study.

2.       Fig2 A is partly covered by Fig2 B. Please check it.

3.       Legend of Fig 2 should be corrected. Some Russian sentences are there.

4.       The letter size of Fig 1 legend should be corrected.

5.       There are some typos throughout the manuscript.

Author Response

Dear reviewer!

The authors thank you for evaluating our article and for comments. We tried to correct the manuscript as much as possible according to your comments.

  1. Data on medication of the enrolled patients should be included. It is well known that some of the drugs used in CAD, including stating may significantly alter the levels of oxidative stress markers and adipokines. This could be also mentioned as a limitation of the study.

Data on medication were added to the Table 1. Unfortunately, we did not find statistical differences between groups in the use of the medications, but, as you can see from table, the differences are present. We added this point to study limitations.

  1. Fig2 A is partly covered by Fig2 B. Please check it.

We checked Fig2

  1. Legend of Fig 2 should be corrected. Some Russian sentences are there.

We apologize for this inconvenience! We have removed our technical notes.

  1. The letter size of Fig 1 legend should be corrected.

We checked Fig 1 legend.

  1. There are some typos throughout the manuscript.

We checked throughout the manuscript and hope that we corrected all the typos.

Reviewer 2 Report

I find the manuscript very interesting and innovative. It is notable for the very good methodological and statistical study of the material studied. However, I have a few minor comments:
1. Subsection: biochemical study - the second and third paragraphs should be shortened and summarised, e.g. that these determinations or the diagnosis of diabetes were based on generally accepted European guidelines.
2. Subsection: Adipose Tissue explants: - the description of the methodology should be shortened.
3. Figure 2 and Figure 3- some of the information in the figure caption is in Russian, not English - should be translated.
4. The manuscript contains too many tables and figures - some of them should be placed like supplementary material e.g. table 7 et etc.
5. The discussion is too long - the first 10 paragraphs are actually an introduction, not a discussion - some of this information could be moved to the Introduction chapter, and the actual discussion could be limited to a discussion of the authors' own findings in relation to the literature data.
6 As the authors also studied adiponectin levels and their correlations with other parameters assessed, the discussion lacks reference to a previously published paper on adiponectin in coronary artery disease - Clin Chim Acta 2012 Apr 11;413(7-8):749-52.doi: 10.1016/j.cca.2012 .01.006. (Adiponectin--an independent marker of coronary artery disease occurrence rather than a degree of its advancement in comparison to the IMT values in peripheral arteries).

Author Response

Dear reviewer!

The authors thank you for evaluating our article and for your comments. We tried to correct the manuscript as much as possible according to your comments.

  1. Subsection: biochemical study - the second and third paragraphs should be shortened and summarised, e.g. that these determinations or the diagnosis of diabetes were based on generally accepted European guidelines.

We shortened paragraph 1 and 3 of biochemical study.

  1. Subsection: Adipose Tissue explants: - the description of the methodology should be shortened.

We shortened Adipose Tissue explants section

  1. Figure 2 and Figure 3- some of the information in the figure caption is in Russian, not English - should be translated.

We apologize for this inconvenience! We have removed our technical notes.

  1. The manuscript contains too many tables and figures - some of them should be placed like supplementary material e.g. table 7 et etc.

We placed table 7 to supplementary materials.

  1. The discussion is too long - the first 10 paragraphs are actually an introduction, not a discussion - some of this information could be moved to the Introduction chapter, and the actual discussion could be limited to a discussion of the authors' own findings in relation to the literature data.

We have shortened Discussion and moved some paragraphs to Introduction.

6 As the authors also studied adiponectin levels and their correlations with other parameters assessed, the discussion lacks reference to a previously published paper on adiponectin in coronary artery disease - Clin Chim Acta 2012 Apr 11;413(7-8):749-52.doi: 10.1016/j.cca.2012 .01.006. (Adiponectin--an independent marker of coronary artery disease occurrence rather than a degree of its advancement in comparison to the IMT values in peripheral arteries).

Thank you very much for your note! We cited this paper (See reference number 44).
